# Analysis of the Effect of Graphene, Metal, and Metal Oxide Transparent Electrodes on the Performance of Organic Optoelectronic Devices

**DOI:** 10.3390/nano13010025

**Published:** 2022-12-21

**Authors:** Ziqiang Chen, Zhenyu Wang, Jintao Wang, Shuming Chen, Buyue Zhang, Ye Li, Long Yuan, Yu Duan

**Affiliations:** 1College of Physics, Changchun University of Science and Technology, Changchun 130013, China; 2State Key Laboratory of Integrated Optoelectronics, College of Electronic Science and Engineering, Jilin University, Changchun 130012, China; 3Key Laboratory of Functional Materials Physics and Chemistry of the Ministry of Education, College of Physics, Jilin Normal University, Changchun 130103, China

**Keywords:** transparent electrodes, organic optoelectronic devices, graphene, ultrathin metal, oxide-metal-oxide structure

## Abstract

Transparent electrodes (TEs) are important components in organic optoelectronic devices. ITO is the mostly applied TE material, which is costly and inferior in mechanical performance, and could not satisfy the versatile need for the next generation of transparent optoelectronic devices. Recently, many new TE materials emerged to try to overcome the deficiency of ITO, including graphene, ultrathin metal, and oxide-metal-oxide structure. By finely control of the fabrication techniques, the main properties of conductivity, transmittance, and mechanical stability, have been studied in the literatures, and their applicability in the potential optoelectronic devices has been reported. Herein, in this work, we summarized the recent progress of the TE materials applied in optoelectronic devices by focusing on the fabrication, properties, such as Graphene, ultra-thin metal film, and metal oxide and performance. The advantages and insufficiencies of these materials as TEs have been summarized and the future development aspects have been pointed out to guide the design and fabrication TE materials in the next generation of transparent optoelectronic devices.

## 1. Introduction

A TE is a transparent conductive film that combines optical transparency and excellent electrical conductivity [1,2,3,4,5,6,7,8,9,10,11,12]. It can be applied to various optoelectronic devices such as flat panel displays, light-emitting diodes, solar cells and smart windows [13,14,15,16,17,18]. The ideal TE has the advantages of high visible light transmission, great conductivity over a large area, and great mechanical flexibility [19,20]. Despite the continued success of TE in the optoelectronics field, for efficient optoelectronic devices, the performance of TEs still requires further improvement. Indium tin oxide (ITO) has high transmittance in the visible wavelength range, and has a very low square resistance (R_s_) at a thickness of around 200 nm [21], therefore, ITO predominates in optoelectronic devices based on rigid substrates [22,23]. In recent decades, the rapid development of optoelectronic devices has led to a growing demand for ITO. However, the high sputtering cost of ITO as well as the low abundance of indium on earth and its toxicity in nature, have limited the application of ITO in optoelectronic devices, meanwhile, and ITO has poor mechanical flexibility, prone to cracking during mechanical strain or bending, which makes it hard to apply to flexible devices [24]. Therefore, how to prepare various types of TEs to replace ITO will facilitate the development of large-area, transparent, flexible and low-cost optoelectronic devices.

In recent decades, several TEs have been developed to replace ITO, including graphene [25,26], metal electrode (ultrathin metal [27], nanowires [28], metal grid [29]), oxide-metal-oxide (OMO) structure [30,31], and conductive polymer [32]. Graphene, a new generation of TEs with high optical transmittance and electrical conductivity, is usually prepared by chemical vapor deposition (CVD), yet it is liable to defects in the preparation process and expensive to prepare [33]. Metal nanowires (MNWs) are prepared by a simple solution method. However, their stability are limited and surface roughness are high [34]. Single carbon nanotube (CNT) has high electrical conductivity, yet the random distribution of CNT-based TEs makes their optoelectronic properties limited [35]. Therefore, MNWs and CNTs are usually prepared by hybridizing with other electrode materials. The metal film is widely utilized in electrodes because of its strong lateral conductivity, yet the thick metal film has high reflectivity due to its own microstructure, hard to be utilized in TEs. Therefore, the metal film is usually prepared with metal oxide to form an OMO structure [36]. OMO structure combines the high electrical conductivity of metal film with the superior optical transmittance of metal oxide, the property of its structure has also made it widely investigated [37].

Herein, this short review systematically introduces how to release the limitations of monolayer electrode. Starting with graphene monolayer, investigating the latest advances in graphene-based TE, including doping effect, optimization of preparation method, and exploration of graphene multilayer. Next, introduce the growth mechanism of ultrathin metal film, and classified in terms of surface free energy (SFE) and preparation method [38,39], including seed layer, dopant metal, and gas. Finally, we introduce OMO-structured TEs, which have both high transmittance and conductivity, and summarize the research on their optical, electrical and mechanical performance. Many reviews on TEs have been published [22,40], differently, we combine our previous work, systematically summarized the doping and modification of TEs monolayer. In order to follow the latest research, we summarized new theory in the past three years, these in-depth research of TEs from different perspectives present novel directions that deserve to be explored.

## 2. Graphene Based TE Materials

Graphene is a two-dimensional (2D) monolayer material with only one carbon atom thickness [41,42]. The most promising method for transparent conductive graphene electrodes in optoelectronic devices is reportedly synthesized via chemical vapor deposition, where graphene films are deposited directly on a substrate such as Cu foil by passing a carbon source through a high-temperature vacuum chamber, and the synthesized graphene is transferred to the target substrate using a support film such as poly (methyl methacrylate) [43,44,45].

### 2.1. Modification and Doping of Graphene

Graphene monolayer is not inherently a highly conductive electrode material, because its defects R_s_ is about 1000 Ω/sq, and the lowest figure of merit (FOM) in the photovoltaic field up to date [46]. As shown in Figure 1a, ideal graphene provides a flawless transport pathway for electrons or holes, avoids the concern of high resistance losses when applied in organic optoelectronic devices. However, graphene monolayer is often produced with defects in the preparation process, as shown in Figure 1b, graphene with defects affects its conductive properties as TE materials, there is a natural effect on the electronic structure, which leads to electron or hole transport is hampered and reduces the lateral conductivity of graphene, and results in low carrier concentration and a large potential energy barrier [47]. Graphene and any other 2-d materials has a very low transmittance and is also a major reason why it’s used in conjugation with other materials and not standalone. Therefore, the modification and stable doping of graphene are extremely important to obtain graphene composite TEs with R_s_ below 100 Ω/sq and above 90% optical transmittance.

#### 2.1.1. Reduced Graphene Oxide

A common modification method for graphene is reduced graphene oxide (RGO), a low-cost method compatible with roll-to-roll (R2R) mass production, where graphene oxide (GO) is chemically peeled by ultrasonic dispersion or rapid thermal expansion, and then reduced by appropriate chemical or laser-assisted methods [48]. RGO can be easily produced in large volumes as graphene ink by taking advantage of its better solubility in common solutions [49]. Based on this situation, RGO TEs have been extensively researched. Without sacrificing the optical transmittance, RGO cannot effectively reduce the resistance and roughness of the bottom layer, therefore, similar to graphene monolayer, RGO also need combine together with other electrode materials [50,51,52].

#### 2.1.2. Composite Preparation of Graphene Transparent Electrodes

The conductivity of graphene monolayer is poor, in order to meet the application of graphene TEs in optoelectronic devices, as shown in Figure 2, graphene monolayer is usually combined together with metal oxide, CNT, MNWs and metal thin film [53,54,55,56].

CNT films have excellent chemical stability and conductivity. However, the adhesion to the substrate is weak, and its unique tube-tube junction can easily open causing a rise in R_s_, so CNT is often hybrid with other electrode materials such as graphene to prepare electrodes. On the basis of this, Geng et al. performed polydopamine (PDA) functionalized of the RGO as plotted in Figure 3a, and thus combined with single-walled CNT (SWCNT) [57]. As illustrated in Figure 3b,c, PDA-RGO/SWCNT/poly(3,4-ethylenedioxythiophene) polystyrene sulfonate (PEDOT:PSS) composite TEs with a structure similar to that of reinforced concrete were fabricated. This structure has excellent mechanical stability during bending, and the PDA-RGO improves the adhesion of the conductive layer with the substrate. The conductivity of the composite electrode is significantly improved, with a sheet resistance of 52.2 Ω/sq and an optical transmission of 88.7% at 550 nm. The composite TE has the potential as an organic light-emitting diode (OLED) anode, and the prepared device has a luminance of 2032 cd/cm^2^ at 15 V and maximum current efficiency of 2.13 cd/A at 14 V.

In our previous work, we provided a graphene optimization method, we prepared Au-doped graphene films, and through improving the concentration of dopant solution (AuCl_3_) and doping time, we prepared flexible transparent graphene electrode with R_s_ reduced from 365 Ω/sq to 90 Ω/sq, and 85.6% transmittance at 550 nm. When applied in OLED device, it exhibited higher performance both in terms of illumination and device efficiency [58]. MNWs are widely applied as TE materials [59]. The distribution of MNWs is random and the conductivity is severely affected by boundary effects, which arise from the fact that the atoms on the surface of MNWs that are not fully bonded as those of the monolithic material, which makes the electrical conductivity of MNWs lower than that of the monolithic material [60,61]. As the size of MNWs decreases, the number of surface atoms increases relative to the number of monolithic atoms, and thus the boundary effects become more pronounced [41]. Good electrical conductivity requires upgrading the length of MNWs. However, micron-scale lengths can make the prepared films have significant protrusions, so MNWs are often hybrid with other electrode materials such as graphene to prepare electrodes.

Accordingly, Grossman et al. reported a highly stable electrophoretic deposition (EPD)-GO/silver nanowires (AgNWs)/GO TE, which has a similar sandwich structure fabricated by utilizing the EPD method for GO, where AgNWs serve as a conductive bridge covered by GO films on both sides [62]. This newly developed all-solution process allowed transparent conductive films to be transferred to alternate surfaces after deposition and demonstrates excellent R_s_ (15 Ω/sq). In addition, simply changing the thickness of EPD-GO, its transmittance can be adjusted from 70% and 87% at 550 nm. In contrast with bare AgNWs, the composite electrode retains its original conductivity over long periods of operation at 80% relative humidity. the AgNWs networks are effectively “interposed” between the two layers of GO providing combined stability and performance.

In a research work published by Berry’s research team, it was demonstrated that ideally flawless graphene can be utilized as a multifunctional encapsulating layer, because the gaps within the aromatic rings of carbon are nearly zero due to the overlapping electron clouds, preventing the penetration of gas molecules, even tiny helium molecules [63].

Accordingly, Grossman et al. utilized ultrathin GO as an encapsulation layer to protect the AgNWs from the PEDOT:PSS-hole transport layer (HTL), thus not affecting the energy level alignment of the device [64]. EPD-GO/AgNWs/GO as an organic solar cell (OSC) anode to substantially improve the lifetime of inverted translucent OSC, without additional encapsulation, the lifetime of the entire device is increased by a factor of five.

#### 2.1.3. ALD-Oxide on Graphene

ALD-oxide (Atomic Layer Depostion-oxide) is an essential material for the perovskite solar cell (PSC) electron transport layer, organic optoelectronic device encapsulation layer, and TEs dielectric layer. However, due to the surface structure of graphene and the step-covering deposition of ALD, ALD oxides are difficult to deposit on graphene, so it is a challenging task to utilize graphene and ALD-oxide simultaneously [34,65,66,67]. Accordingly, as shown in Figure 4, people usually deposited a functional layer before depositing the ALD-oxide on the graphene, which can be a useful aid to the deposition of the ALD-oxide [68]. Herein, we utilized ethylene glycol (EG) as a precursor to prepare a functional layer to activate the graphene surface and thus improve the deposition of ALD-ZnO films [69]. The prepared EG-Graphene/ALD-ZnO composite films exhibit a significant reduction in R_s_, higher transmittance, smooth and uniform surface, excellent bending ability and energy level matching with other device components. Lee et al. utilized NO_2_ to provide nucleation sites for the deposition of ALD-Al_2_O_3_ on graphene to increase surface reactivity, promoting in the deposition of uniform Al_2_O_3_ layers on graphene [70].

### 2.2. Other Methods

Contrary to doping and modification, graphene monolayer can also be prepared without composite with other conductive materials. For instance, adhesion enhancement between the graphene monolayer and substrate compensates the defects of the graphene monolayer by graphene multilayer.

Graphene monolayer has poor adhesion to the substrate, and the graphene surface is easily contaminated by dust during the process of peeling off from the metal foil or transferring to the target substrate. These challenges make it difficult to utilize in industrial production. Koo et al. combined graphene film directly with polyimide (PI) to obtain TEs with high flexibility and thermal stability [71]. As plotted in Figure 5, the PI containing dual functionalities on CVD-grown graphene, which acts as both a carrier film and a substrate for the graphene electrode, exhibits an ultra-clean surface, along with an optical transmittance of over 92%, a sheet resistance of 83 Ω/sq. In addition, the direct integration of PI improves the durability of the graphene electrode by reducing the mechanical exfoliation process of the graphene. The power conversion efficiency (PCE) of OSC has obtained significant progress of 15.2%.

The photovoltaic properties of graphene multilayer, as compared to graphene monolayer, have also been extensively researched. However, contrary to the main theme of this review, therefore, it will not be described in detail.

## 3. Ultrathin Metal Transparent Electrode

Ultrathin metal film is utilized in TEs one step earlier than ITO and act as a more viable replacement for ITO. It features a powerful charge transfer capability, a smooth surface, can be prepared by simple preparation processes such as thermal evaporation or sputtering, no conflicts with functional layers, and stability different from metal grids and MNWs.

Due to the surface conditions of the substrate and the variety of metal film, the deposition of thin metal film usually follows three various modes [72]. The Volmer-Weber (island growth) mode occurs when the interaction between the proximity metal atoms is stronger than the interaction between the substrate and metal atoms, as shown in Figure 6a, the metal atoms are bonded with each other and then with the substrate surface atoms. Instead, the Frank-Vander Merwe (layer growth) mode occurs when the interaction between the substrate and metal atoms more strongly than the interaction between the proximity metal atoms, in which case the film follows a strictly 2D growth mode, as shown in Figure 6b, this means that the metal atoms bond with the atoms on the substrate first, one layer is completely grown before the next layer starts to grow. The third growth mode between the above two is the Stranski-Krastanov mode (island/laminar growth), when one or two monolayers of material are first deposited and then individual islands are deposited on it [73]. The growth of ultrathin metal film usually conforms to the Volmer-Weber method, starting with individual islands of metal at 10–20 nm, they later converge to form a continuous conductive film.

Although ultrathin meatal film can achieve high transmittance, the Volmer-Weber mode results in non-continuous and rough films with poor electrical conductivity. Through studying the process of metal film formation, it can be observed that the metal has a certain micro-structure on its surface at a thin thickness. This micro-structure not only enhances the scattering of light at the interface, but also is not conducive to the transmittance of transverse currents, and is detrimental to the optical and electrical properties of the thin metal film. Therefore, the deposition of the thin metal film is usually facilitated by increasing the SFE of the substrate or the binding energy between the metal and the substrate, resulting in an ultrathin and sleek continuous metal film, which are important for TEs applied to ITO-free optoelectronic devices [74].

Ag and Au films are often utilized in ultrathin metal TEs due to their lowest intrinsic resistivity [75]. However, metal materials have high reflectivity due to a large number of free electrons. Consequently, homogeneous metal layers generally exhibit low optical transmittance, so the main goal of the research about ultrathin metal TEs is to without complete loss of its electrical conductivity, making the metal transparent. In the subsequent, we summarize three methods to obtain transparent and well-conductive Ag and Au ultrathin metal TEs.

### 3.1. Increasing the Surface Free Energy of the Substrate

The Volmer-Weber mode is mainly due to an energy imbalance between the metal and the substrate, where the high energy of the metal surface leads to strong interaction effects between metal nanoparticles rather than with the substrate, resulting in a rough metal film. Therefore, increasing SFE of the substrate is an effective strategy to enhance the performance of TEs [76].

Building on this, a variety of methods have been explored, such as growing a seed layer before growing the metal film, oxide treatment of the substrate surface [77], etc. Herein, this review focuses on varieties of seed layer nanomaterials that are effective as an ultrathin metal TE for OLED or PSC devices, such as: molecule, metal, polymer, graphene, and metal oxide. Their structures are all shown in Figure 7 [78,79,80,81,82].

The most commonly available seed layer material is metal, such as Cr [83,84], Ge [85], Au [86], and Cu [87,88], due to their high surface energy, they have strong adhesion to commonly used Ag and Au atoms, and inhibit the spread of precious metal clusters. Priya et al. used the Cr seed layer to prepare ultrathin Au film as semi-transparent PSC top electrode, the highest PCE reached 19.8%, and has high transmittance in the near- infrared wavelength range [83]. Song et al. formed continuous ultrathin film by introducing a 1 nm Cu seed layer, which resulted in a low percolation threshold of 5 nm Ag electrodes, which not only ensured good conductivity but also broadband transparency. The results showed a PCE of 14.5% for semi-transparent PSC using a 6 nm Ag electrode, maintaining 88% of the performance of non-transparent PSC [87]. Moreover, the optical transmittance, surface roughness, and lateral conductivity of the transparent metal electrode based on the metal seed layer are summarized in Table 1.

As mentioned in the previous section, the FOM of graphene monolayer is low, it usually prepared by doping or modifying to enhance the electrode performance. However, it can also be utilized as a seed layer for ultrathin metal films. In this work, Sun et al. used the graphene monolayer as a seed layer for 7 nm ultrathin Au film and prepared composite TEs, showed an average R_s_ as low as 24.6 Ω/sq and 74.6% transmittance at 520 nm [81]. The capacitance of the device prepared with this composite TE was increased by a factor of 17 compared to the device prepared with a single layer of graphene. Our group combined graphene monolayer with 8 nm Ag film, showed a low R_s_ of 8.5 Ω/, 74% transmittance at 550 nm, and high stability over 500 bending cycles [94].

Compared to the metal seed layer, polymer as seed layer eliminated the high reflectivity of the metal itself, Kang et al. used polyethyleneimine (PEI) as a seed layer for ultrathin Ag films on flexible poly (ethylene glycol)2, 6-naphthalate (PEN) substrates [80]. The amino group of PEI provides an alone-pair electron to the Ag atom thus forming a coordination bond, where Ag can better bond with the substrate rather than Ag to Ag. This allows for the formation of uniform, aggregated Ag nuclei on the substrate surface, resulting in the subsequent formation of high-quality, ultra-thin Ag films. Furthermore, by solution treatment, the surface energy difference is formed and the charge is redistributed, which reduces the work function of the composite TE from 5.1 eV to 3.9 eV, established favorable ohmic contacts for efficient inverted PSC device. Sun et al. used polyamide-imide [poly(trimellitic anhydride chloride-co-4,4′-methylenedianiline)] (PAI) as a seed layer for ultrathin Ag electrode and MoO_3_ as a reflectance-reducing layer to increase the transmittance of the composite TE, the PAI/Ag/MoO_3_ composite electrode has a low R_s_ of 15.1 Ω/sq, high transmittance of 87.4% and a surface roughness of 0.768 nm [95].

Molecule seed layers avoid the high time cost of polymer seed layers and the optical degradation of the metal film, usually spin-coating or depositing a molecular layer film on the substrate can provide nucleation sites for the deposition of metal atoms and prevent their island growth mode [96,97]. By utilizing molecular layer deposition (MLD), in our previous work, we have prepared methylated alucone film as a seed layer for ultrathin Au electrodes, increasing the SFE of the substrate to aid nucleation of Au electrodes [98]. The growth illustration of alucone film is shown in Figure 8. This results in high optoelectronic performance and a bending life of more than 1500 times. This demonstrates the prospect of ultrathin Au electrodes replacing ITO electrodes in organic optoelectronic devices.

It is worth mentioning that we also prepared PMMA/trimethylaluminum (TMA) via low-temperature ALD half-reaction, the methyl provided by TMA is well applied as seed layer for ultrathin Au TE [99]. It not only improved by the optical transmittance and lateral conductivity of ultrathin Au TE, but the seed layer also reacted with PMMA substrate, then formed a self-encapsulation layer, which greatly improved the stability and lifetime of the device. Based on the optoelectronic synergistic Ca test, its water vapor transmittance rate is as low as 2.123 × 10^−3^ g/cm^2^/d (60 °C/80% RH), which greatly improved the stability and lifetime of the OLED device prepared by this ultrathin Au electrode [17]. Furthermore, at a 1 mm bending radius, this device can still maintain its original performance. In addition, the performance of transparent metal electrodes based on polymer and molecular seed layers is summarized in Table 2.

The metal oxide is commonly utilized as a seed layer in OMO structures, and their specific applications will be elaborated on in the next section.

It can be found that the transmittance of ultrathin metal TEs based on the seed layer acquires a large percentage increase in the long wavelength range, which is because the seed layer eliminates the photothermal effect of the metal material when the metal absorbs light of longer wavelength, this part of light has little energy and cannot cause electronic excitation, which mainly makes the molecules and atoms of the irradiated material move faster and produce the thermal effect. The smooth and continuous film reduces the heat generated by the light being absorbed by the material, thus increasing its transmittance.

In general, research in this area is in process, because metal atoms as seed layer increase the optical absorption of TEs thus reducing the transmittance. The wettability of dielectrics is limited, the preparation of graphene is more difficult, the preparation of polymers is usually not suitable for low vacuum, and the surface molecular structures prepared by ALD are still in the laboratory stage. With joint efforts, all of the preparation methods may be applied in industrialization in the future.

### 3.2. Reduce the Interfacial Free Energy between the Metal and the Substrate

Doping metal into the host metal through a co-deposition process is the second approach for inhibiting the metal Volmer-Weber mechanism, which is to reduce the interfacial free energy between the metal and the substrate, without the application of a seed layer and critical fabrication conditions. The doped metal is usually chosen to have greater binding energy to the substrate surface atoms, which binds to the substrate surface atoms one step earlier compared to the host metal, provides a nucleation site and improves the uniformity of the film formation [108]. If the substrate surface is an oxide or other material, a metal that binds better to the substrate surface atoms can also be chosen. Guo and his coworkers, who co-deposited a small amount of Al during Ag deposition, reported an ultrathin and smooth Ag film that showed low optical loss and low resistance. This method did without a seed layer, and the Al-doped Ag film showed enhanced PCE for organic photovoltaics fabricated on such slim walled electrodes due to light trapping within the photoactive layer [109]. In addition, the devices exhibit significant bending capability and stability compared to ITO-based devices. In their subsequent studies, it was found that Cu-doped Ag would be more suitable compared to Al, because Cu has higher binding energy to the substrate or oxide [110].

### 3.3. Reduce the Surface Free Energy of Metal

Doping gas into the deposited metal through a co-deposition process is the third approach for inhibiting the metal Volmer-Weber mechanism, which is to reduce the SFE of metal, without the application of a seed layer and critical fabrication conditions, and mainly affects the early growth stages of metals, mainly about O and N. It is reported that the SFE of O-doped Cu can be reduced by nearly 60% compared Cu [111]. However, the control of O dose is a critical issue, to avoid the formation of oxide phase. This may cause dramatic destruction of the electrical conductivity of the metal film, which is challenging for R2R mass production. Likewise, Ni-doped Cu is a direct approach to avoid the formation of the oxide phase. In the early growth stages of Cu film, doping Ni can effectively inhibit the Volmer-Weber mode and cluster behavior of Cu [112].

## 4. Oxide–Metal–Oxide Structure

OMO electrodes combine the high conductivity of metal with the high transmittance of oxide [113,114]. The optoelectronic performance of OMO electrodes has been reported to significantly exceed that of ITO. As mentioned in the previous section, the bottom oxide layer functions as a seed layer and enhances the film-forming ability of the metal film [115], and the charge redistribution with the metal layer and the Schottky barrier reduction effect, there will be a high charge carrier concentration in the OMO structure. The wrapping of the metal film by the top layer of oxide makes the stability of the electrode much higher than that of a monolayer metal film, and also allows the selection of the appropriate oxide layer according to the application requirements of the device [116]. More strategically, the top oxide layer reduces the reflection and increases the transmittance of the metal film, and its structural characteristics are illustrated in Figure 9. Meanwhile, the dense oxide layer can act as a self-encapsulation layer compared to the ultrathin metal TE and increase the device lifetime [99]. Therefore, this section provides an overview of OMO electrodes in three parts: optical, electrical and mechanical stability.

### 4.1. Optical

Due to the high reflectivity of metal materials, TEs based on metal materials exhibit very low transmittance, and the oxide in the bottom and top layers of the OMO structure helps to improve the transmittance of the electrode in the visible and near-infrared light range. Based on the previous doped metal TE, Guo and his coworkers prepared a ZnO/Cu-doped Ag/Al_2_O_3_ structure of OMO TE using a small amount of Cu-doped Ag [117]. The relative transmittance >100% is achieved by quantitative analysis to reduce the reflection of metal films and increase the optical transmittance due to the optimized OMO electrode structure, and the replacement of the top layer of oxide material allows the electrode to be applied in optoelectronic devices.

Optical simulations provide important guidance for determining the optimal thickness of each layer of the OMO structure. For instance, finite-difference time-domain (FDTD), transport matrix method, etc. [37]. Lee and coworkers measured the film thickness, refractive index, and extinction coefficient of the OMO electrode and performed optical simulations with Essential Macleod, and changed the thickness of the bottom and top oxide layers to increase the transmittance while keeping the R_s_ constant, and similar to the experimentally measured data, the optimized SiInZnO (SIZO)/Ag/SIZO TE has 96.67% transmittance at 550 nm and R_s_ is 6.41 Ω/sq [118].

### 4.2. Electrical

Research shows that the oxide layer has a strong influence on the metal electrode, metal in contact with oxide can exhibit rebalancing of the electron, giving the oxide the electrical conductivity of the metal. Compared to monolayer metals, OMO electrodes exhibit changes in conductivity and work function. Meanwhile, the oxide layer has better interfacial modification than the metal layer for the whole device. However, most of the research on OMO electrodes has focused on their optical transmittance, and neglected the electric charge transfer effect induced by the metal layer. Herein, we deposited flawless ZnO nanointerfaces on Ag layer by ALD, the combination of ZnO with Ag made the prepared surface ZnO exhibit metallic behavior, and improved the interfacial stability of ZnO/Ag/ZnO (ZAZ) electrode. More noteworthy, degenerated state of the ZnO exhibited a change in the work function. This discovery enabled better energy level alignment when applied in organic optoelectronic devices [119] Compared to conventional ITO electrodes, this metal electrode shows better performance in both OLED and perovskite light-emitting diode devices. Based on ZAZ, Kim et al. prepared Ni-doped ZAZ (Ni:ZAZ) as flexible blue OLED anodes for thermally activated delayed fluorescence (TADF) using co-sputtering [120]. The composite electrode work function is adjustable and has excellent optoelectronic performance, the R_s_ is 6.33 Ω/sq and the transmittance is 93.3% at 484 nm.

The conductivity of OMO structures is usually influenced by the film formation quality of the metal film, so many researchers are currently addressing this problem by improving its growth behavior during the deposition of Ag [121]. Yun et al. prepared highly smooth, less defective, fully continuous 4.5 nm ultrathin Ag film by adding O atoms to the outermost layer of Ag film to inhibit island growth, which can be attributed to the reduction of metal SFE by O atoms [122]. During the growth process, the existence of O atoms inhibits the production of the Ag oxide phase and the strong structural integrity of the ultrathin Ag(O) film is successfully maintained, as plotted in Figure 10a. Ultrathin Ag(O) film sandwiched between ZnO films with an OMO structure can maintain their conductivity at an annealing temperature of 673 K. It far exceeds the temperature limits of sterling Ag, and also exhibits wide transmittance in the overall visible light range of 400–800 nm. The film-forming ability of Ag can also be enhanced by chemisorption [79,107]. Jen et al. covalently bonded functional-group-terminated molecular monolayer surfactants (FMMSs) containing thiol end groups and carboxylic acid end groups to Ag and ZnO, respectively, and improved the interfacial adhesion of Ag to ZnO with sheet resistance less than 10 Ω/sq [104].

### 4.3. Mechanical Property

Due to the technological development of flexible optoelectronic devices, the long-term mechanical stability of TEs is very important. Different from ITO electrode, OMO electrodes are robust because of their middle metal layer. In our previous bending experiments, we observed that as the bending radius decreased, the OMO electrode and ITO exhibited a great difference under the optical microscope, ITO electrode surface appeared a large number of cracks, while only small cracks were observed in OMO electrode, because the metal layer is located on the neutral axis of the electrode and its bending performance is only influenced by its top and bottom oxide layers, whose modulus and thickness are minimal. In this recent work, the internal mechanism of flexible electrodes was well researched, Guo and his coworker presented the failure sequence of OMO electrodes under internal and external bending is delamination, buckling, and channel cracking [123]. As illustrated in Figure 10b, which compared the crack propagation modes of monolayer ITO and ITO/Cu:Ag/ITO, thus exhibiting the non-conductivity of ITO and the conductivity of ITO/Cu:Ag/ITO. In general, under bending condition, due to the difference in Young’s modulus between the oxide and the polymer substrate, the oxide film is often accompanied by microscopic cracks, and when a thin metal layer was deposited on the oxide, its good ductility improved the electrode bending resistance.

Song et al. prepared cellophane/ZAZ TE by adjusting the thickness of the substrate cellophane [124]. It has a R_s_ of 7.2 Ω/sq and an average 81.7% transmittance at 400–800 nm. Next, perform neutral axis calculations to reconcile the functional layer of PSC to the neutral axis. PSC has good mechanical properties and can maintain 98% of the initial value after 500 bending cycles at a 1 mm bending radius and 92% of the initial value after 35 folding cycles.

MoO_3_ is a suitable material for hole injection layer, Ag or other metals sandwiched between MoO_3_ layers are currently applied in organic optoelectronic devices [125,126,127]. However, the mechanical properties are still in the research stage. Kim et al. reported the effect of the thickness of Ag on the mechanical stability of MoO_3_/Ag/MoO_3_ (MAM) structures [128]. The optimal thickness of the Ag layer in MAM was identified by experimentally calculating the highest FOM, and obtained a R_s_ of 3.17 Ω/sq and 77.85% optical transmittance. Next, prepare MAM electrode on colorless PI, by performing outer bending, inner bending, twisting, and rolling test methods to indicate excellent mechanical stability. In addition, the optoelectronic and bending performance of TEs based on different OMO structures are summarized in Table 3.

## 5. Conclusions

Graphene, ultrathin metal, and OMO structure had attracted strong research interest as a new generation of TEs for rigid and flexible ITO-free organic optoelectronic devices. The optical transmittance and lateral conductivity of TEs significantly affected the performance of optoelectronic devices. Whether graphene, metal, or the single layer film preparation process inevitably produced defects, its optoelectronic performance is limited. However, such limitations can be released by doping and modification. This short review systematically introduced the latest progress in enhancing the optoelectronic properties of these three TEs through doping and modification. All improved TEs had the advantages of low roughness, high transmittance, excellent lateral conductivity, and the preparation process met the requirements of industrialization, now or in the future can be well applied in the industrial production of optoelectronic devices. This mainly includes poor conductivity of graphene monolayer with high transmittance, a problem solved by modification, doping with other conductive materials and other preparation methods. The Volmer-Weber growth mode of ultrathin metal resulted in poor film formation quality, inhibited by seed layer and doping, and classified by SFE mechanisms. Due to OMO structural features, introducing the optimization of its optical, electrical and mechanical stability achieved high quality, surface-continuous TEs, successfully balancing the ideal value of optical transmittance and lateral conductivity. Based on this, ITO-free organic optoelectronic devices based on graphene, ultrathin metal, and OMO structures exhibited excellent performance, with good efficiency and mechanical stability.

## Figures and Tables

**Figure 1 nanomaterials-13-00025-f001:**
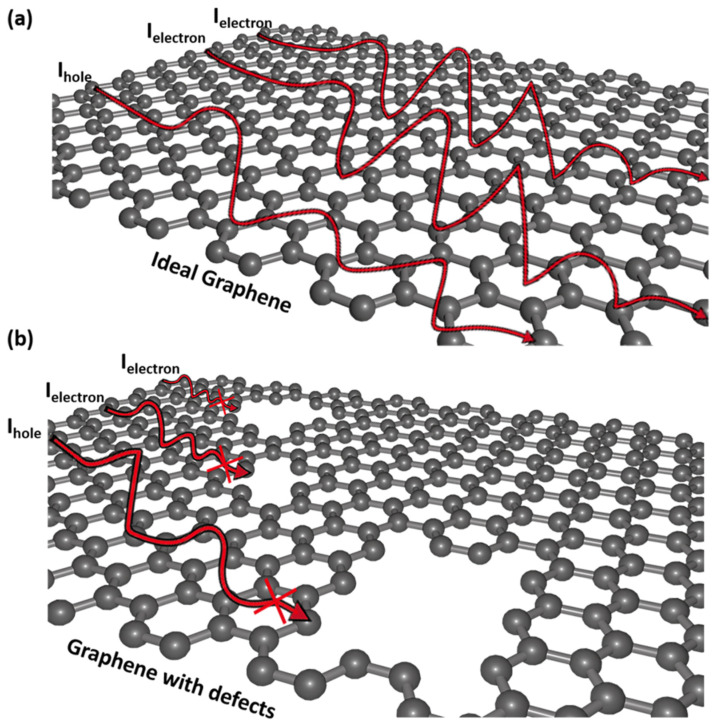
The schematic illustration of electron or hole transport path in: (**a**) ideal graphene provides a flawless transport pathway for electrons or holes, while it is hampered in (**b**) defective graphene.

**Figure 2 nanomaterials-13-00025-f002:**
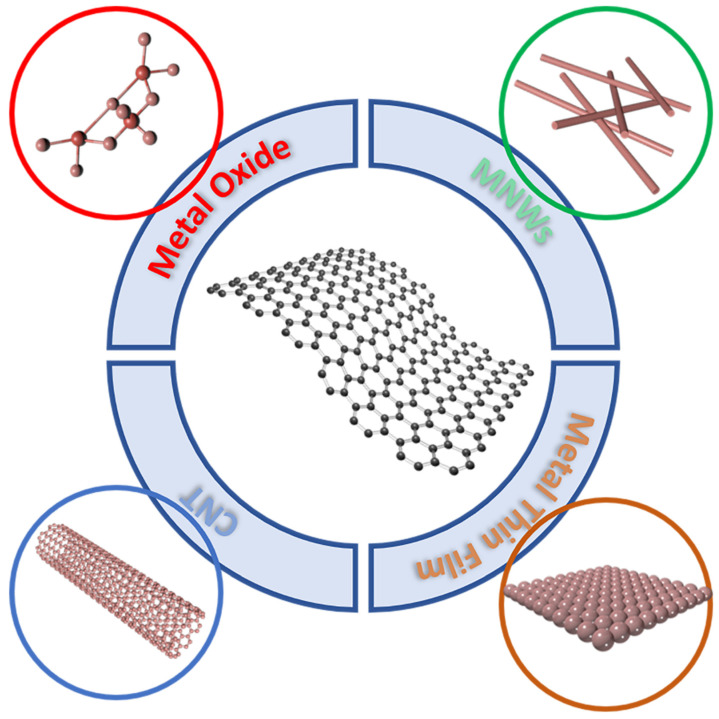
The schematic illustration of graphene composite preparation with typical nanomaterials. Such as metal oxide, MNWs, CNT and metal thin film, resolved the poor lateral conductivity of graphene monolayer.

**Figure 3 nanomaterials-13-00025-f003:**
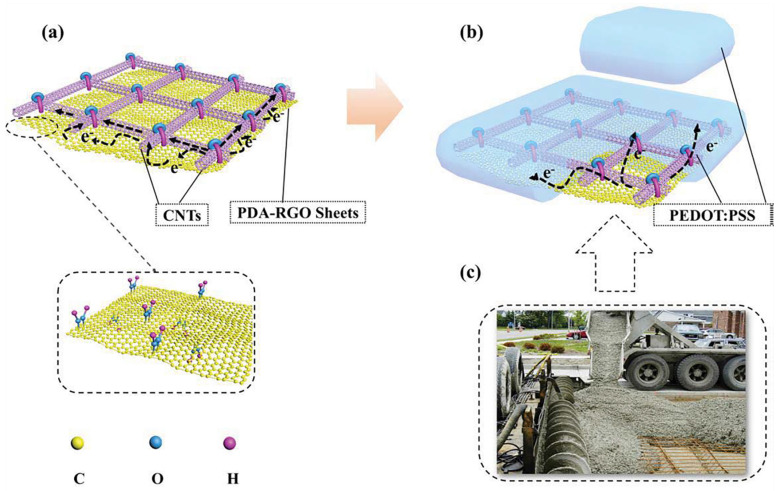
The schematic illustration of TCFs with ultra-adhesion, relatively low roughness, and excellent electrical conductivity: (**a**) PDA−RGO/CNT and (**b**) PDA−RGO/CNT/PEDOT:PSS. (**c**) A picture of reinforced concrete structure used in building construction. (Reproduced from [57] with permission from Elsevier, 2020).

**Figure 4 nanomaterials-13-00025-f004:**
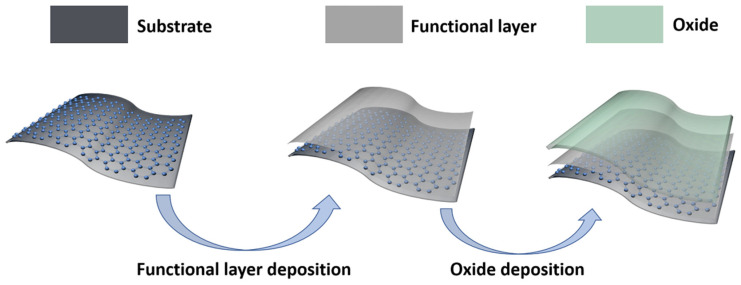
The solution for the deposition of oxides on flexible graphene substrate. The schematics, respectively, illustrate the flexible substrate with graphene, the deposition of a functional layer on graphene, and the fact that the oxide can be deposited well after the deposition of the functional layer.

**Figure 5 nanomaterials-13-00025-f005:**
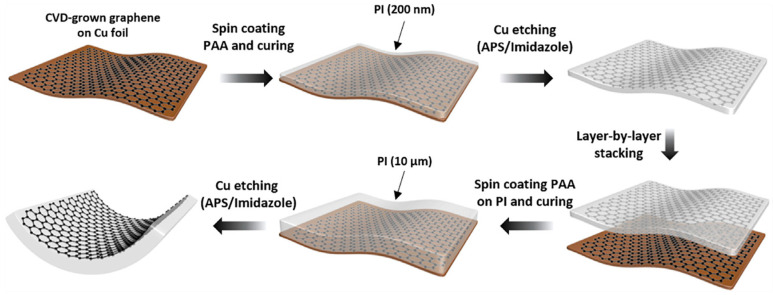
200-nm-thin PI and 10-mm-thick PI were applied as a carrier film during graphene transfer and a substrate for the graphene electrode, respectively (Reproduced from [71] with permission from Elsevier, 2020).

**Figure 6 nanomaterials-13-00025-f006:**
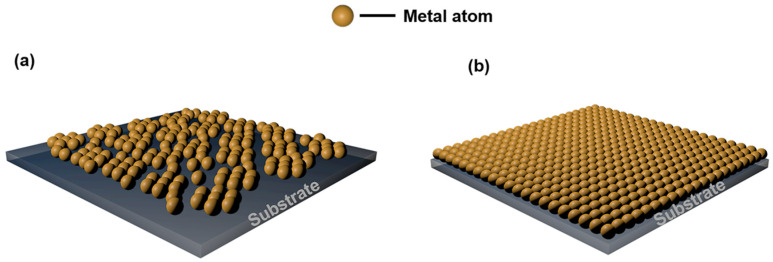
Two various growth modes that result in a metal film appearance. A schematic illustrating (**a**) Volmer-Weber mode and (**b**) Frank-Vander Merwe mode.

**Figure 7 nanomaterials-13-00025-f007:**
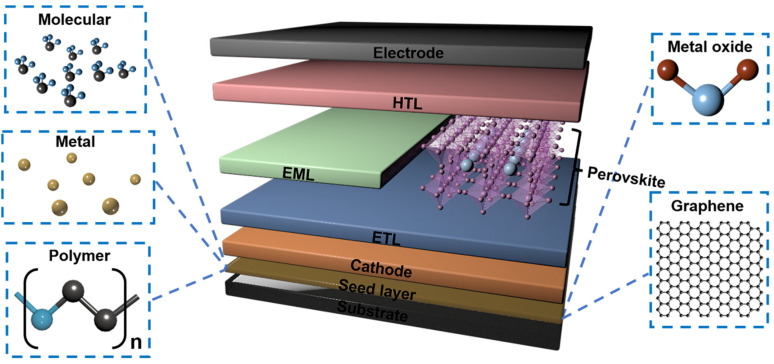
The schematic illustration of the classification of seed layers that are effective as an ultrathin metal TE for OLED or PSC devices, including molecule, metal, polymer, metal oxide and Graphene.

**Figure 8 nanomaterials-13-00025-f008:**
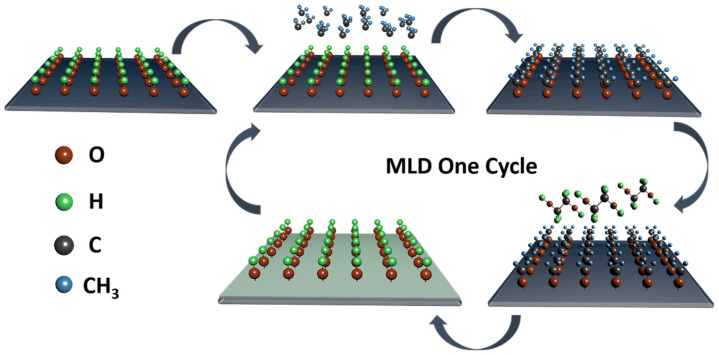
The schematic illustrating MLD-alucone film one cycle of growth process.

**Figure 9 nanomaterials-13-00025-f009:**
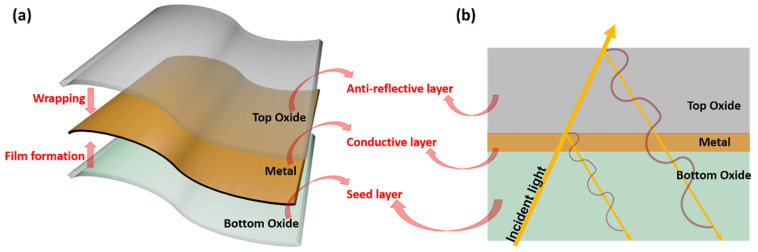
The schematic illustration of each layer of OMO electrode. (**a**) The features of each layer and (**b**) destructive interference in the OMO film with anti-reflective layer.

**Figure 10 nanomaterials-13-00025-f010:**
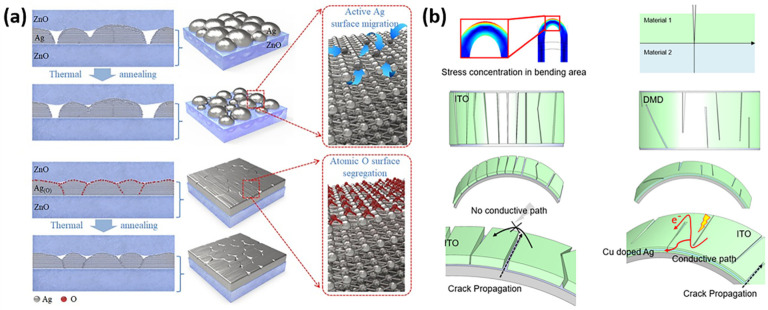
(**a**) Atomic O-induced film formation of Ag. The schematic comparison of the film formation of Ag and Ag(O) between the top and bottom ZnO dielectric layer. (Reproduced from [122] with permission from Elsevier, 2021). (**b**) The finite element analysis indicated that the cracks generated by the fracture would appear in the bending area, strain is proportional to the sample thickness, and inversely proportional to the bend radius. Furthermore, compared the crack propagation modes of monolayer ITO and ITO/Cu:Ag/ITO, thus exhibiting the non-conductivity of ITO and the conductivity of ITO/Cu:Ag/ITO. (Reproduced from [123] with permission from American Chemical Society, 2022).

**Table 1 nanomaterials-13-00025-t001:** A summary of transmittance, roughness and R_s_ of transparent metal electrodes based on metal seed layer.

Seed Layer	Metal	Method	Transmittance	Roughness (nm)	R_s_ (Ω/sq)	Device Application	Ref
Cr	Au	evaporation	≈70% @350–1200 nm	-	16.3	PSC	[83]
Cr	Au	evaporation	60%	-	27.9	OSC	[84]
Ge	Ag	evaporation	85.33% @550 nm	0.11–0.16	25	-	[85]
Ge	Ag	evaporation	-	0.4	14.01 ± 1.4	-	[89]
Au	Ag	evaporation	85% @550 nm	-	16	PSC	[86]
Ag	Au	evaporation	80%	-	16	Flexible OLED	[90]
Ni	Ag	evaporation	75%	3.9	11	flexible OSC	[91]
Al	Ag	evaporation	87% @550 nm	-	19.5	OLED	[92]
Cu	Ag	evaporation	62.77% @400–1200 nm	0.204	19	semi-transparent PSC	[87]
Cu	Au	evaporation	75% @550 nm	5.4	16	PSC	[93]

**Table 2 nanomaterials-13-00025-t002:** The summary of transmittance, roughness, and R_s_ of transparent metal electrodes based on polymer and molecular seed layer.

Seed Layer	Metal	Metal Deposition	Transmittance	Roughness (nm)	R_s_ (Ω/sq)	Device Application	Ref
PEI	Ag	evaporation	≈80% @550 nm	0.23	≈9	PSC	[80]
PAI	Ag	evaporation	87.4% @550 nm	0.768	15.1	flexible OLED	[95]
PEI	Ag	evaporation	69.7% @550 nm	<1	6.3	semi-transparent OSC	[100]
PVK	Ag	evaporation	>85%	-	<10	various optoelectronic devices	[101]
SU-8	Au	evaporation	72% @550 nm	0.35	23.75	OLED	[102]
SU-8	Au	-	≈80% @550 nm	0.575	19	flexible OSC	[103]
MUA	Ag	evaporation	78% @400 nm	0.95	13.59	flexible OSC	[104]
PEI	Ag	evaporation	80% @550 nm	0.15	9	flexible LED	[105]
PAA	Ag	evaporation	75% @550 nm	-	10	flexible LED	[106]
PVP	Ag	evaporation	70% @550 nm	-	19	flexible LED	[106]
PMMA	Ag	evaporation	45% @550 nm	-	∞	flexible LED	[106]
PFN	Ag	evaporation	54.3%	1.3	9.4	semi-transparent polymer solar cells	[106]
Alucone	Au	evaporation	77.8%	0.18	30	OLED	[98]
PMMA/TMA	Au	evaporation	84.25% @550 nm	0.566	18.19	flexible OLED	[99]
MPTMS	Ag	evaporation	76%	0.5	6	semitransparent and flexible PSC	[96]
MPTMS/APTMS	Au	evaporation	>80% @495–600 nm	0.4	11	organic photovoltaic	[97]
(3-aminopropyl) trimethoxysilane	Au	evaporation	-	0.19	-	-	[79]
(3-mercaptopropyl) trimethoxysilane	Au	evaporation	75% @550 nm	0.2	-	-	[107]

**Table 3 nanomaterials-13-00025-t003:** A summary of transmittance, roughness, R_s_ and flexibility of various OMO structure.

Structure	Metal Deposition	Transmittance	Roughness (nm)	R_s_ (Ω/sq)	Flexibility	Device Application	Ref
ZnO/Cu:Ag/Al_2_O_3_	sputter	100.3% @300–700 nm	<1	≈18.6	-	-	[117]
SIZO/Ag/SIZO	sputter	96.67% @550 nm	-	6.41	-	-	[118]
ZnO/Ag/ZnO	evaporation	80% nm @450 nm	≈0.3	2.6	-	OLED	[119]
Ni:ZnO/Ag/ZnO	evaporation	93.3% @484 nm		6.33		flexible OLED	[120]
ZnO/Ag(O)/ZnO	sputter	85% @550 nm	0.47	-	Bending radius 4 mm	-	[122]
ZnO/Ag/ZnO	evaporation	85% @550 nm	0.95	8.61	-	flexible organic photov-oltaic cell	[104]
Cellphone/ZnO/Ag/ZnO	sputter	91% @400–800 nm	-	7.2	35 folding cycles	foldable PSC	[124]
MoO_3_/Ca:Ag/MoO_3_	evaporation	95% @550 nm	-	27.1	Bending cycles 800	flexible OLED	[127]
MoO_3_/Ag/MoO_3_	evaporation	77.85%	-	3.17	Various test methods	-	[128]
WO_3_/Ag/WO_3_	evaporation	≈90%	0.72	9	Bending cycles 2000	flexible photov-oltaic cell	[129]
SnO_x_/Ag/SnO_x_	evaporation	82%	-	9	Bending radius 5 mm	OLED	[130]

## Data Availability

All data is contained within the manuscript.

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
