# Peer review of "Analysis of the Effect of Graphene, Metal, and Metal Oxide Transparent Electrodes on the Performance of Organic Optoelectronic Devices"

_nanomaterials, 2022, doi:10.3390/nano13010025_

Round 1

Reviewer 1 Report

Report on the review: “Analysis of the effect of graphene, metal, and metal oxide transparent electrodes on the performance of organic optoelectronic devices.”

“A transparent electrode (TE) is a transparent conductive film that combines optical transparency and excellent electrical conductivity. [1,2]” For a review, 2 references are not enough to account for all the different possible solutions to replace ITO. Therefore, it seems necessary to recall the properties of the metal oxides the most often used today and to recall some previous reviews dedicated to possible substituent to TCO:

Many reviews have been already published such as:

-Recent progress in transparent oxide semiconductors: materials and application, H. Hosono, Thin Solid Films 515 (2007) 6000 – 6014.

-Discovery-based design of transparent conducting oxide films, G. J. Exarhos, X-D Zhou, Thin Solid Films 515 (2007) 7025 – 7052.

-Transparent conductors as solar energy materials: A panoramic review, C. G. Granqvist, Solar Energy Materials & Solar Cells 91 (2007) 1529 – 1598.

- Transparent conducting oxides for photovoltaics, A. E. Delahoy and S. Guo, in: Handbook of Photovoltaic Science and Engineering, edited by A. Luque and S. Hegedus (John Wiley & Sons, Ltd., New York, 2011), pp. 716–795.

- K. Arapov, A. Goryachev, G. de With, H. Friedrich, A simple and flexible route to large-area conductive transparent graphene thin-films, Synth. Met. 201 (2015) 67-75.

- C. Guillén, J. Herrero, TCO/metal/TCO structures for energy and flexible electronics, Thin Solid Films 520 (2011) 1-17.

- L. Cattin, J.C. Bern_ede, M. Morsli, Toward indium-free optoelectronic devices: dielectric/Metal/Dielectric alternative conductive transparent electrode in organic photovoltaic cells, Phys. Status Solidi A 210 (2013) 1047-1061.

-R. Po, C. Carbonera, A. Bernardi, F. Tinti, and N. Camaioni, Polymer and carbon-based electrodes for polymer solar cells: Toward low-cost, continuous fabrication over large area, Sol. Energy Mater. Sol. Cells 100, (2012) 97 - 114.

-Transparent conducting oxides for electrode applications in light emitting and absorbing devices, H. Liu, V. Avrutin, N. Izyumskaya, Ü. Özgür, H. Morkoç, Superlattices and Microstructures 48 (2010) 458 – 484.

-Emerging transparent conducting electrodes for organic light emitting diodes, T-B. Song, N. Li, Electronics 3 (2014) 190 – 204.

About the manuscript itself, the authors chose to present first graphene, then thin layers of metals and finally OMO structures.

Concerning grapheme, it is well known that if theoretically its resistivity is very small, experimentally graphene monolayers are produced with defects, which limits significantly their conductivity. One possibility consists in doping graphene with conductive bridges of CNT, MNWs and oxides or/and to use composites with polymer such as PEDOT:PSS. Then ultra thin metal films used as transparent electrodes are presented. Here it is necessary to find the best compromise between transparency and conductivity, since the thicker the layer, the higher is its conductivity but the less transparent it is. Unfortunately, the growth of ultrathin metal film usually starts with individual islands of metal at 10-20 nm, which means that relatively thick films are necessary to form a continuous conductive film. An increase of the SFE of the substrate allows obtaining continuous metal layer for smaller thickness. Such STE increase can be achieved through substrate surface treatment, growing a seed layer before growing the metal film, reducing the interfacial free energy between the metal and the substrate.

Another possibility consists in using Metal/Oxide/Metal (OMO) structures, which combine the high conductivity of metal with the high transmittance of oxide. Here, the bottom oxide layer functions as a seed layer, while the top layer of oxide makes the stability of the electrode higher than that of a monolayer of metal film. Moreover it is possible to choose the oxide layer according to the application requirements of the device. It must be noted that the oxide in the bottom and top layers of the OMO structure improves the transmission of the electrode in the visible. Moreover, these OMO electrodes are flexible.

This work deserves interest, it is a good presentation of current research dedicated to new transparent electrodes.

Before publication, acronyms must be introduced: “ALD”-Oxide on Graphene, “Gr” and references must be completed.

Reviewer 2 Report

I consider that the paper do not add major contribution to the already existing books or review scientific articles on this topic.

Reviewer 3 Report

In this review article, authors have summarized the transparent electrodes (TEs) based on graphene, thin metal films, and metal film in conjugation with the metal oxides. The article will serve as a good reference for the scientific community who are working on the development of the transparent electrode as an alternative to ITO. Reviewer supports the publication with following recommendations.

·         Author should highlight in their introduction how their review is different compared to other published work on transparent electrodes for organic optoelectronics devices, couple of examples, of many other reviews available online

o   https://doi.org/10.1117/1.JPE.4.040990

o   https://pubs.rsc.org/en/content/articlelanding/2019/tc/c8tc04423f

·         Graphene and any other 2-d materials has a very low transmittance and is also a major reason why it’s used in conjugation with other materials and not standalone, thye should highlight it

·         Authors title and main claim is to describe the effect on the performance of the organic optoelectronic devices. However, the review mainly focuses on the material and growth aspect of the TEs and not significantly on the comparison of how the performance was improved w.r.t. reference by employing those TEs

o   They should summarize and tabularize the performance data of the optoelectronic devices where such TEs were used and weather they are used as anode cathode, etc. instead of just putting the transmittance and resistance data

o   They should also put add column mentioning how the optoelectronic devices were prepared by solution or vacuum based methods

·         It’s not just about the transmittance  and resistance but work function of the electrodes matters the most for the performance of the organic or any optoelectronic devices, they should add some comments or section on that in the review.
